# Heteroplasmy and Individual Mitogene Pools: Characteristics and Potential Roles in Ecological Studies

**DOI:** 10.3390/biology12111452

**Published:** 2023-11-20

**Authors:** Wenhui Wang, Lijun Lin, Qi Zhang, Jincheng Yang, Elizabeth Kamili, Jianing Chu, Xiaoda Li, Shuhui Yang, Yanchun Xu

**Affiliations:** College of Wildlife and Protected Area, Northeast Forestry University, Harbin 150040, China; 18435173810@nefu.edu.cn (W.W.); monkkiiidddd@gmail.com (L.L.); 15735171383@163.com (Q.Z.); maturecheng@163.com (J.Y.); elizabeth.mtui@mwekawildlife.ac.tz (E.K.); 17614666298@163.com (J.C.); lixiaoda99@163.com (X.L.)

**Keywords:** MtDNA heteroplasmy, individual mitogene pool, molecular marker

## Abstract

**Simple Summary:**

The mitochondrial genome is a multicopy circular DNA with high mutation rates due to replication and repair errors. Mitochondrial DNA (mtDNA) heteroplasmy refers to the presence of more than one type of mtDNA in an individual. A mitochondrion, cell, tissue, organ, or an individual body may hold multiple variants, both inherited and developed over a lifetime. We review the characteristics, dynamics, and functions of mtDNA heteroplasmy and propose the concept of an individual mitogene pool to discuss individual genetic diversity from multiple angles to guide how the individual mitogene pool with novel genetic markers can be applied to ecological research.

**Abstract:**

The mitochondrial genome (mitogenome or mtDNA), the extrachromosomal genome, is a multicopy circular DNA with high mutation rates due to replication and repair errors. A mitochondrion, cell, tissue, organ, or an individual body may hold multiple variants, both inherited and developed over a lifetime, which make up individual mitogene pools. This phenomenon is also called mtDNA heteroplasmy. MtDNA variants influence cellular and tissular functions and are consequently subjected to selection. Although it has long been recognized that only inheritable germline heteroplasmies have evolutionary significance, non-inheritable somatic heteroplasmies have been overlooked since they directly affect individual fitness and thus indirectly affect the fate of heritable germline variants. This review focuses on the characteristics, dynamics, and functions of mtDNA heteroplasmy and proposes the concept of individual mitogene pools to discuss individual genetic diversity from multiple angles. We provide a unique perspective on the relationship between individual genetic diversity and heritable genetic diversity and guide how the individual mitogene pool with novel genetic markers can be applied to ecological research.

## 1. Introduction

Recent research advances on mitochondrial genomes (mitogenomes) are attributed to the advent of sequencing technologies and the simultaneous integration of multiple disciplines. The earliest breakthrough was in 1982, when the base substitution patterns of mtDNA sequences from different species were used to infer their evolutionary relationship [1]. By the end of the 20th century, whole and partial mitogenomes had been used to evaluate genetic diversity, estimate gene flow between populations, infer differentiation and phylogenetic relationships between populations and taxonomic units, reconstruct evolutionary history, identify maternal lineages and species, as well as to study the genetic basis of mitochondrial function and diseases [2].

In addition to the significance of the mitochondrion as a power house in eukaryotic cells, the genetic functions of mitogenomes include DNA replication and transcription, RNA processing and translation, and post-translation maturation of newly synthesized polypeptides. Each cell contains numerous mitochondria, and each mitochondrion contains multiple copies of the mitogenome. Prior to the detection of the simultaneous presence of two mtDNA haplotypes, termed “mtDNA heteroplasmy”, in a *Drosophila mauritiana* [3], it was widely accepted that the mitogenome in all cells of an organism was identical in sequence. Technically, heteroplasmy refers to the presence of more than one type of mitochondrial DNA (mtDNA) in an individual [4]. Heteroplasmy is a copy-level mitogenome variation, and each cell, tissue, organ, and individual can be a reservoir of various variants [5]. The root causes of mitogenome heteroplasmy include erroneous replication produced by the low fidelity of DNA polymerase [6] in germinal and somatic cells; high levels of reactive oxygen species (ROS) in cells, leading to mtDNA damage [7]; and subsequent repair errors during DNA replication [8]. Mutations occurring at a nucleotide position are a random process and independent from each other, while mutation patterns along the mitogenome may show distinct individuality at all levels, from cells to organs, individuals, populations, and even species [9].

Empirical studies show that both wild-type and mutant copies of the mitogenome serve as transcriptional templates when mitochondrial genes are expressed [10], and the post-transcriptional pools of mRNA, tRNA, and translated proteins are mixtures of variants which manifest a diverse set of different sequences, conformations, and physiological efficacies [11,12,13]. Subsequently, this diversity will supposedly prompt deviations from the normal biochemical and physiological efficiency of mitochondria, cells, tissues, and organs. The frequency changes in different variants of wild-type and mutant copies of mtDNA may depict different fitness levels, which may result in a widespread genetic load that affects individual fitness [14]. In addition, heteroplasmic mutations in oocytes may be transmitted down to offspring and become a source of genetic diversity [15] after a sharp reduction due to a severe bottleneck [16].

The dynamics of mtDNA heteroplasmy are modulated by different intrinsic and extrinsic factors. Aging and toxic environmental factors affect heteroplasmy genesis, and organisms gain tolerance to heteroplasmy by homogenizing the mitochondrial gene pool through fission and fusion processes [17]. Meanwhile, when heteroplasmy severely influences function, organisms remove dysfunctional mitochondria through autophagy [18]. Thus, mitogenome heteroplasmy lies in generation and elimination dynamics and is closely related to the environment, senescence, disease, and other factors. This provides a novel approach to understanding the populations’ fitness across species in a particular eco-environment from an mtGenomic perspective.

Over time, different techniques have been employed to measure the dynamics of heteroplasmy. The main drawback of conventional techniques of quantifying heteroplasmy, such as polymerase chain reaction (PCR) and Sanger sequencing, is the failure to detect heteroplasmic base substitutions at rates of less than 10–15% [13]. Such restriction has been overcome by the use of next-generation sequencing (NGS) technology, with significant improvements in detection range, throughput, and sensitivity to characterize heteroplasmies as low as 1.14% in frequency on a genome [19]. 

Compared to chromosomal DNA, mtDNA is easier to obtain from a variety of biomaterials, including muscle, blood, bone, keratinized tissues, secretions of animals, callus cultures, leaf tissues of plants [20,21], fungal spores, and mycelia of fungi [22], allowing people to study heteroplasmy for all of these organisms. Accumulating reports on humans, model animals, and some wildlife taxa provide deepening insights into the mechanisms and consequences of heteroplasmy. Understanding these may open a new window into an mtGenomic basis for broad insights into ontogenetic processes, tissue function, individual fitness, and the evolution of wildlife. In this review, we attempt to generalize the phenomena, characteristics, effects, and dynamics of mitogenome heteroplasmy from published findings and provide perspectives on their ecological significance and potential application in ecological research.

## 2. Characteristics of mtGenomic Heteroplasmy

Heteroplasmy is bound to erroneous replication in all tissues, organs, and throughout developmental stages [23,24] and generally characterized by distinctive patterns. In terms of time, heteroplasmy denotes the frequency change in a given variant from oogenesis to embryogenesis and later in life [25]. Because each copy of mtDNA acts as a template for replication (akin to asexual reproduction), variants generated in the early stages are likely to become more widely distributed in tissues and organs than those that occur later [26]. Spatially, it is evident that mtDNA heteroplasmy varies from cell to cell, tissue to tissue, and organ to organ over time [27,28] and depicts hierarchical mosaicism across these levels (Figure 1A). 

Secondly, mtDNA heteroplasmy is always in a state of genesis-elimination dynamics. Literally, the mitogenome replication presents a chance for heteroplasmy due to polymerase error [6], likely to occur at any base position across the mitogenome. However, Gu and his colleagues highlighted the occurrence of directional selection against substitutions at some points or regions, with the strength decreasing from the first to the third codon [29]. This phenomenon is consistent with directional selection in natural populations that eliminate mutations at certain sites while reserving them at others. This leads to a significant variation in heteroplasmic levels across segments of the mitogenome. It is well known that, at the population level, the nucleotide polymorphism of the mitogenome is the most pronounced in non-coding regions, followed by non-protein-coding regions (rRNA, tRNA genes), and uncommonly in protein-coding regions [10]. Likewise, at the tissue level within a body (Figure 1B) [25,27], heteroplasmic mutations are less frequently eliminated in non-coding regions than in coding ones [30]. The directional selection determining a variant’s fate arises from tissue functions [31], implying that genomic diversity evolves within a tissue.

Thirdly, it has been established that heteroplasmy is directly proportional to age. For instance, in humans, the number of heteroplasmic sites in skeletal muscle accumulates over a lifetime with a linear correlation with age in years (r = 0.746 at the 0.01 significance level) [30]. In addition, variants derived from the original oocyte may persist in all derived cell populations, while later variants occurring during ontogenesis affect only specific body parts [32]. Coherently, the rapid accumulation of specific heteroplasmic nucleotide polymorphisms and total heteroplasmy provides insights into age-related diseases [33,34,35]. The progressive accumulation of heteroplasmy may also lead to tissue senescence [36], and in turn, the senescence could impair cellular rehabilitative capacity and further improve heteroplasmy levels [37].

## 3. The Influence of Heteroplasmy on Cellular Functions

The influences of heteroplasmic nucleotide substitutions on cellular functions vary greatly depending on the mtGenomic region in which they occur. For non-coding regions, the conformational stability of core elements required for replication is not sensitive to nucleotide substitutions [38], while for coding regions the consequences are significant. The RNA coding region encodes 22 transfer RNAs (one tRNA per amino acid, with two for serine and leucine) and 2 ribosomal RNAs (12S and 16S rRNA). Heteroplasmic substitutions may alter the melting temperature, conformation, and stability of tRNAs and may further lead to translational and respiratory chain defects, manifesting as an alteration of membrane potential, reduced proton electrochemical potential gradient, and reduced ATP production [11,12,13]. A vigorous example is that motility and fertilization ability vary significantly among spermatozoa ejaculated at the same time in domestic fowl [39], caused by the A-to-G transition of the tRNA^Arg^ gene at position 11,177, resulting in excessive Ca^2+^ uptake in sperm with low motility [40].

The protein-coding region encodes 13 polypeptides of four respiratory chain complexes in mammals. The changes in amino acid sequence by nucleotide substitutions may lead to structural and functional changes in respiratory chain complexes I and III [41,42], with the possibility of a large proportion impairing the oxidative phosphorylation (OXPHOS) system and ultimately increasing the production of reactive oxygen species [43]. Once this happens, an excessive ROS concentration and dysfunctional OXPHOS system induce the accumulation of pyruvate, CoA, and other basic metabolites [44]. This stimulates the production of co-transcriptional regulators PGC-1α and other transcription factors like HIF1 and NRF1 [43,45] and finally enhances the production of mitochondrial transcription factor A (TFAM), which stimulates mitochondrial biogenesis [46,47]. Furthermore, oxidative damage increases the affinity of the D-loop to TFAM to stimulate mtDNA replication and transcription and boosts mitochondrial biogenesis [35]. Mitochondrial biogenesis entails extensive replication of the mitogenome, which may increase the number of mutations, in turn affecting the sequence of the peptides (Figure 2).

The effects of mtDNA heteroplasmy are solely based on experimental findings using alternative variants of mitogenome. However, it is rational to expect that the effects of heteroplasmic variants in tissues are dose-dependent and that the functional reduction in the entire OXPHOS system may not be significant, provided that the proportion of deleterious variants does not exceed a certain threshold [48]. Generally, above an optimal threshold, germline heteroplasmy may affect the function of cellular tissues and even the body. For example, some mitochondrial diseases are caused by mutations or heteroplasmy of mtDNA in humans (Table 1). However, the number of such heteroplasmic variants in the body is often very few. Most heteroplasmic variants occur during development and aging. There are a minority of mitogenomes in a cell compared to germline haplotypes, each making slight impacts on mitochondrial function, but the overall effects cannot be neglected given the proportion that exceed the threshold [49]. Therefore, the level and number of variants of heteroplasmy, together with heterogeneity among organs, may be potential indicators of individual physiological performance in relation to fitness in a given ecological environment.

## 4. Dynamics of Heteroplasmy

### 4.1. Generation of Variants during Ontogenesis

If an individual animal is considered a mitogene pool, consisting of a number of heteroplasmic variants, the development of the mitogene pool involves the generation and elimination of variants during the ontogenic process. The primary mode of variant generation is de novo erroneous replication, which depends on the fidelity of Polγ, the sole DNA polymerase in mitochondria responsible for all synthetic reactions in mitogenome replication, repair, and recombination [71]. Compared to the nuclear DNA polymerases, whose average replicational error rate is about one error in 10^−6^ to 10^−8^ base pairs [23,24], Polγ has a lower fidelity and error-correcting ability, with an average error rate of around ~3 × 10^5^ base pairs [72]. Thus, mitogenome replication is prone to de novo mutation [6,73]. Variants can also arise secondarily as a result of erroneous repair of ROS-induced damage to de novo variants. Because the mitochondrial matrix is proximate to the respiratory chain, mitochondrial DNA is vulnerable to ROS damages, leading to a single-strand break, double-strand break, interstrand crosslink, and base oxidation, such as converting the nucleotide guanosine to 8-oxo-2′-deoxyguanosine (8-oxo-dG) [74]. Due to the low fidelity of Polγ, erroneous nucleotides can be added to the strand during replication through strand breaks and crosslinks [75]. In this instance, unlike the normal pairing between dG and dC, the 8-oxo-dG pairs with adenine (dA). This mismatch leads to a G: C to T: A transversion in subsequent rounds of replication [75,76,77]. 

Both ROS-induced and Polγ-induced substitutions happen on a small proportion of mitogenomes with tissue dependence [30]. These substitutions show a striking bias in strand orientation: T→C/G→A transitions occur preferentially on the light strand, whereas C→T/A→G occurs on the heavy strand. Substitutions on the heavy chain are more prominent than on the light chain because the majority of mitogenome progeny is derived from the heavy strand after multiple rounds of DNA replication [78].

### 4.2. Tolerance to Heteroplasmy and Elimination

Heteroplasmic variants can be neutral, advantageous, or deleterious, but the majority are thought to be deleterious [18,79]. However, they generally take a small proportion of the mitogene pool, posing a challenge for cells and tissues to maintain homeostasis. The cells have evolved a counteracting mechanism called mitochondrial fission and fusion, which allow two mitochondria to fuse, pooling organellar materials, including mitogenomes, and then splitting into a homogenized variants mixture [17]. In this manner, heteroplasmic variants can spread from one mitochondrion to the entire mitochondrial population in a cell and become diluted by the majority of normal haplotype(s) at the same time to prevent deviation from normal cell functioning [80]. 

However, when dilution cannot prevent deleterious effects of variants, the malfunctioning mitochondria and associated variants are likely to be removed by selective mitophagy [81,82,83]. Additionally, some heteroplasmic variants containing special sequences may have replication problems and finally disappear from the mitogene pool of a cell [84]. Besides in situ elimination, a proportion of heteroplasmic variants in the maternal germline may not be recruited to the oocyte due to the drift during cytogenesis and lose the chance to transmit to the next generation [85]. 

### 4.3. Dynamics of Heteroplasmy in an Individual and across Generations

Contrary to the conventional assumption of population genetic diversity that the mtDNA haplotype is identical along a maternal line and that each individual animal is considered monohaplotypic, the heteroplasmy-based concept of the mitogene pool addresses genomic diversity and its dynamics. As generalized in Figure 3, an ovum ready for fertilization may contain up to 100,000 copies of mitogenomes with a few variants, and so is a zygote. During its development into an individual, variants may spread to different organs and may also disappear. At the same time, de novo variants may arise as a result of massive mitogenome replications. As a result, the newly formed individual mitogene pool expands progressively. In the ovary, each primordial germ cell may produce ~200 copies of a mitogenome, with only a tiny fraction of variants having a chance to be recruited to the oocyte [85,86], showing a drastic drift [87,88,89] whereby the profile of selected variants in a mature ovum may differ substantially from each other so that the mitotic pool of the resulting individual can be seeded differently. 

However, the empirical evidence suggests that the drift is not a completely random process. A study in humans shows that some minor and deleterious variants can be preferentially propagated and finally make it to the next generation regardless of their original frequency [90]. This suggests that variants are involved in function-based selection, and consequently, the composition of offspring’s mitogene pools is difficult to predict from the oocyte profile of the variants. 

## 5. Perspectives for Wildlife Research

The mitogenomes act as blueprints for replication and transcription; therefore, they are structurally and functionally conservative across taxa [27]. Meanwhile, the mechanisms of replication, damage, and repair are all conservative [91]. Most of the available heteroplasmic data and findings are derived from research on humans and model animals. It is reasonable to expect that these characteristics of heteroplasmy are also applicable to wildlife across taxa, at least for mammals. Thus, the individual mitogene pool has strong potential to explore the novel biological and ecological characteristics of wildlife.

Three feasible dimensions for studies of individual mitogene pools in wildlife are the landscape of heteroplasmy; spatial–temporal dynamics; and the relationship between the mitogene pool and fitness. The fundamental dimension is the landscape of heteroplasmy on a mitogenome, focusing on the degree of variation in heteroplasmy by identifying the hot regions (highly heteroplasmic) and cold regions (with less or no heteroplasmy) in a species or higher taxa. The second dimension is the spatial and temporal dynamics of the landscape that address the variation in the mitogene pool among cell subtypes, tissues and organs in an individual animal, and their dynamics at different life stages. The third dimension articulates the relationship between the mitogene pool and fitness traits, focusing on causal factors. 

Generally, fitness assesses an individual’s advantage and disadvantage in traits for survival and reproduction in an environment [92]. Notably, the environment contextual fitness of an individual and a population can be re-defined in terms of a genetic basis linked with a gene pool, and not with traditional linkage with a haplotype [93]. This novel definition allows for the incorporation of the spatial and temporal dimensions of fitness in response to environmental factors. Although the mitogene pool studied here is of somatic cells and not inheritable, it influences the chances of oocyte mitogenome variants to descend to the next generation by influencing the fitness of their carriers, and has evolutionary significance. 

Despite its technical feasibility, the dynamics of the mitogene pool in wildlife are yet to be explored. The mitogene pool is essentially the heteroplasmy. It is accurately detectable for variants and their proportions by using NGS technology. NGS sequences each DNA template individually with dramatic precision for heteroplasmy detection [94]. It adapts to sequence both the PCR amplicons of a target region and the entire mitogenome [95]. Nowadays, with the increasing fidelity of DNA polymerases, the PCR-induced error rate is drastically reduced, and the detection sensitivity is greatly improved [96]. PCR-free NGS completely avoids such errors by using deep sequencing, e.g., 1000× for nucleotide heteroplasmy below 3% and 3000× for those below 1.5% [97]. A PacBio-based single-molecule sequencing method has been established to detect heteroplasmy with a sensitivity of 1% and support characterizing variants at haplotype levels with long reads [98]. Meanwhile, developing bioinformatics tools and accumulating well-vetted data and nuclear mitochondrial fragment (NUMT) data progressively improve the accuracy and sensitivity of heteroplasmy detection [99,100,101]. Meanwhile, the heteroplasmy could be recorded on mtDNA of all biomaterials, including live tissues and dead hair shafts [102,103]. Non-invasive samples such as feces, hair, feathers and reptilian sloughs, etc., are readily available and feasible for analysis using the current advent technologies to facilitate wildlife studies from the perspective of the mitogene pool.

## 6. Conclusions and Perspectives

MtDNA heteroplasmy arises from the replication and erroneous repair of the mitogenome throughout the whole life history of an organism. It exists at multiple levels, from a single mitochondrion to cells and tissues, constituting the individual mitogene pool with landscape heterogeneity among organs and across ages. Heteroplasmic variants can be neutral, advantageous, and deleterious and influence the functions of cells and tissues. Organisms buffer their effects by mitochondrial fusion and fission and eliminate heteroplasmic variants through the autophagy of mitochondria and cells when they exceed thresholds. Such dynamics are shaped by intrinsic factors such as aging and extrinsic factors such as exposure to DNA-damaging substances. This characteristic establishes its potential to be a novel tool to study interactions between organisms and their environments. With the continuous advancement of next-generation sequencing and single-molecule sequencing technologies, mtDNA heteroplasmy could be widely applied to wildlife studies to either address old questions or raise new questions from the angle of the individual mitogene pool that cannot be reached by traditional approaches.

## Figures and Tables

**Figure 1 biology-12-01452-f001:**
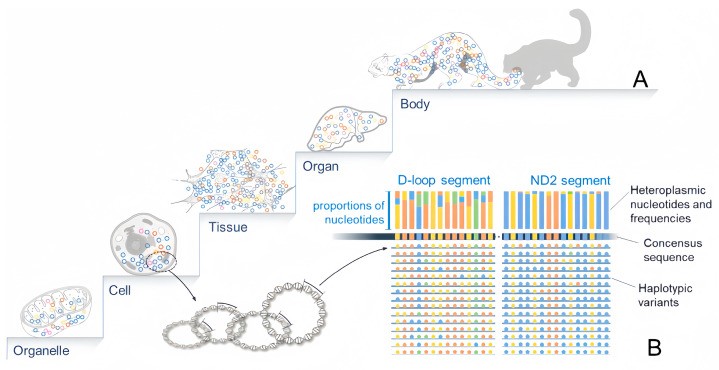
Hierarchical mosaicism of mtDNA heteroplasmy and nonrandom distribution across nucleotide positions and tissues. (**A**). All hierarchical levels of life structure, from organelle (mitochondrion), cell, tissue, and organ to the whole body, may contain heteroplasmic variants of mtDNA and show mosaic patterns. (**B**). Diagram showing the heteroplasmy of a segment of the mitogenome. Multiple copies of mtDNA from a cell or tissue consist of multiple variants revealing polymorphisms at nucleotide sites. The consensus sequence of this fragment is determined by the major nucleotide at each site, while heteroplasmy is revealed by minor nucleotides. The level of heteroplasmy varies from region to region on the mitogenome, highest in D-loop and considerably lower in other regions such as ND_2_.

**Figure 2 biology-12-01452-f002:**
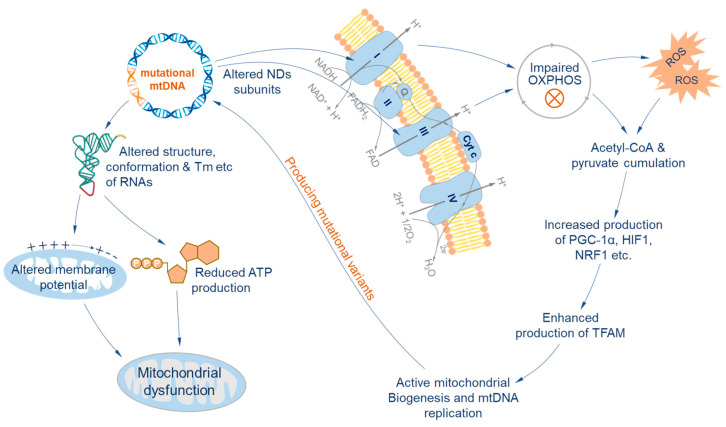
Effects of mtDNA heteroplasmy. Mutations in RNA coding regions may alter the structure, conformation, and melting temperature of transfer RNAs and ribosomal RNAs, and further lead to alteration of membrane potential and reduction in ATP synthesis [11,12,13]. These changes all result in the dysfunction of mitochondria. Meanwhile, mutational variants in protein-coding regions may alter the structure and function of NDs subunits, further impairing the function of the OXPHOS system. Impaired OXPHOS will lead to the accumulation of ROS, acetyl-CoA, pyruvate, and other basic metabolites, and further lead to hypoxia of cells. The hypoxic state of cells stimulates the production of co-transcriptional regulators PGC-1α and other transcription factors like HIF1 and NRF1 and finally enhances the production of mitochondrial transcription factor A (TFAM) that stimulates mitochondrial biogenesis and mtDNA replication. Active mtDNA replication in turn facilitates the production of additional mutational variants. OXPHOS, oxidative phosphorylation; PGC-1α, peroxisome proliferator-activated receptor-gamma coactivator 1alpha; HIF1, hypoxia-inducible factor 1-alpha; NRF1, nuclear respiratory factor 1; TFAM, mitochondrial transcription factor A.

**Figure 3 biology-12-01452-f003:**
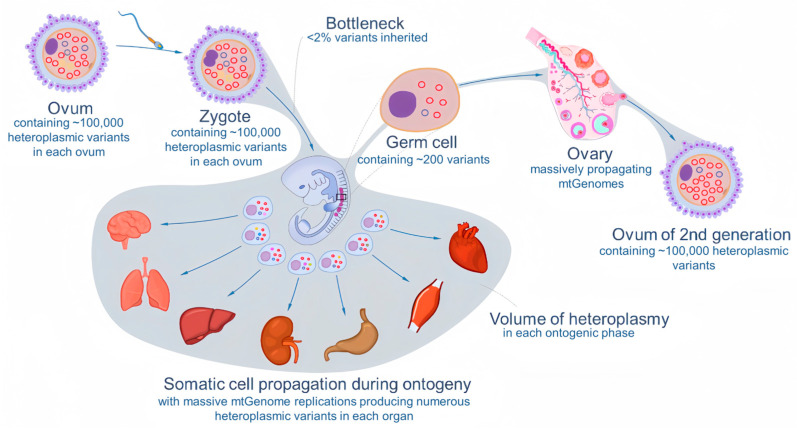
Heteroplasmy passing through bottleneck and development of individual mitogene pool. Each ovum contains up to 100,000 copies of mitogenomes with a few variants. The zygote develops into an embryo with a set of tissues and organs. This process involves massive cell proliferation and replication of the mitogenome, and numerous de novo variants are produced. The de novo and inherited variants form a mitogene pool for each organ and the whole body. Each of the primordial germ cells derived from a group of embryonic stem cells inherit a tiny fraction of the mitogene pool of the entire germline through a bottleneck and drastic drift. During the ovum’s development and maturation, the inherited mitogenomes are massively propagated, producing a large number of de novo heteroplasmic variants available to second-generation animals. Nevertheless, de novo variants are always low-frequency, making the majority of variant(s) in an organ the ones inherited from the previous generation.

**Table 1 biology-12-01452-t001:** Mitochondrial diseases caused by mtDNA mutations.

Mitochondrial Diseases	Mutation Point	Gene	Reference
Deafness	m.1494C>T, m.1555A>G	12S rRNA	Ding et al., 2016 [50], Xiang et al., 2019 [51], Dai et al., 2019 [52]
Deafness	m.1555A>G, m.3243A>G, m.7511T>C, m.7445A>G	tRNA^Val^, tRNA^Leu(UUR)^, tRNA^Ser(UCN)^	Mutai et al., 2017 [53], Finsterer and Zarrouk-Mahjoub, 2018 [54]
Mitochondrial Cardiomyopathy	m.4300A>G, m.3250T>C, m.3303C>T	tRNA^Ile^, tRNA^Leu(UUR)^	Finsterer, 2023 [55], Perli et al., 2012 [56], Hu et al., 2020 [57]
Myopathy	m.5703G>A, m.4308G>A, m.14674T>C, m.14709T>C, m.3302A>G	tRNA^Asn^, tRNA^Ile^, tRNA^Glu^, tRNA^Lam(CUN)^, tRNA^Leu(UUR)^, tRNA^Ser(UCN)^	Fu et al., 2019 [58], Souilem et al., 2011 [59], Mimaki et al., 2010 [60], Mezghani et al., 2010 [61], Ballhausen et al., 2010 [62]
MERRF	m.8344A>G, m.8356T>C	tRNA^His^, tRNA^Lys^	Blakely et al., 2014 [63], Nakamura et al., 2010 [64],
CPEO	m.4308G>A	tRN^Ile^	Souilem et al., 2011 [59]
Leigh’s Disease	m.10158T>C, m.10191T>C, m.10197G>A	ND3, ND4, ND5, ND6	Finsterer and Zarrouk-Mahjoub, 2017 [65], Ahmadi et al., 2022 [66], Hechmi et al., 2022 [67]
LHOP	m.3460G>A, m.11778G>A, m.14484T>C	ND1, ND4, ND6	Yu et al., 2010a [68], Jiang et al., 2016 [69], Yu et al., 2010b [70]

Note: MERRF: Myoclonic Epilepsy with Ragged-Red; CPEO: Chronic Progressive External Ophthalmoplegia; LHOP: Leber’s Hereditary Optic Neuropathy Fibers.

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
