# Peer review of "Heteroplasmy and Individual Mitogene Pools: Characteristics and Potential Roles in Ecological Studies"

_biology, 2023, doi:10.3390/biology12111452_

Round 1

Reviewer 1 Report

Comments and Suggestions for Authors

Reviewer's report 

Date: 16 October, 2023 

Journal: Biology

Type of manuscript: Review

Title: Heteroplasmy and Individual Mitogene Pools: Characteristics and Potential Role in Ecological Studies

Authors: Wenhui Wang, Lijun Lin, Qi Zhang, Jincheng Yang, Elizabeth Kamili, Jianing Chu, Xiaoda Li, Shuhui Yang *, Yanchun Xu *

This review focuses on the characteristics, dynamics, and functions of mtDNA heteroplasmy. The authors propose a new concept concerning individual mitogene pool. 

The manuscript is clear and well written, with no fundamental flaws and weaknesses, and contains new and interesting data that are sound, adequately described and illustrated, and that may provide important cues to scientists interested in thereby support the usage of mitochondrial DNA in biological and medical studies. Therefore, after amending the manuscript according to the following mostly technical suggestion, it is suitable for publication in Biology. 

Minor points: 

Line 48: The abbreviation “mtgenome” is better to change to “mitogenome”.  It should be done here and until the end of the manuscript (e.g., lines 52, 79, 93, 145, 208, Figure 3, 319). 

Lines 57-59: The sentence is not completely clear. Consider reformulation. 

Lines 110-114: These two sentences need to be clarified. Directional selection supports particular genetic variant (Directional Darwinian Selection). Purifying selection by definition cannot “reserves” genetic variability. Purifying selection selects against particular variant. 

Figure 1: The quality of the figure is low. Increase resolution and size of the figure. 

Figure 2: It is unclear where the beginning and the end of this drawing is. Please add the appropriate designations.

Line 187: The references [11-13] are in bold face. Change. 

Line 238: Use Italics for “in situ”. 

Line 248: Use Italics for “de novo”.

Lines 256-257: “A study in humans shows some minor and deleterious variants can be preferentially propagated…” Explain the mechanism. Why “deleterious variants can be preferentially propagated…”? 

Figure 3: In the figure caption, the authors talk that each ovum contains about 100,000 copies of mitogenomes, and in the figure itself, the authors show that the ovum about 100,000 heteroplasmic variants. Clarify the inconsistency. 

Line 266: Change “primodial” to primordial. 

Lines 269, 270: Use Italics for “de novo”.

Line 302: Transfer the abbreviation (NGS) to the page 2, line 84, where you use it first. 

Line 383: Reference # 19: Add the source. 

Reviewer 2 Report

Comments and Suggestions for Authors

The manuscript “Heteroplasmy and Individual Mitogene Pools: Characteristics and Potential Role in Ecological Studies” defines the role of mtDNA heteroplasmy in cellular function, aging, and organism diversity. This manuscript highlights the important role of heteroplasmy in various biological functions. The manuscript is well written and might attract wider attention in scientific society. However, I have a few suggestions regarding the manuscript:

1.       Please add a section and a figure depicting the importance and stages of mitophagy and mtDNA heteroplasmy.

2.       Although most of the mutations involved in mtDNA heteroplasmy are recessive, many human disorders have been associated with it. Please add a section/table correlating human disease and mtDNA heteroplasmy.

3.       Although the manuscript is well written the critical evaluation of the scientific findings is missing. Please add your critical evaluation wherever necessary in the manuscript.

Round 2

Reviewer 2 Report

Comments and Suggestions for Authors

The authors have addressed the issues raised in the first review to best of their abilities. So, I am satisfied with this manuscript for further processing. Thank you.